# Bayesian assessment of chlorofluorocarbon (CFC), hydrochlorofluorocarbon (HCFC) and halon banks suggest large reservoirs still present in old equipment

**Megan Jeramaz Lickley**[1], **John S. Daniel**[2], **Eric L. Fleming**[3,4], **Stefan Reimann**[5], and **Susan Solomon**[1]

[1]Department of Earth, Atmospheric and Planetary Sciences, Massachusetts Institute of Technology, Cambridge, MA 02139, USA
[2]NOAA Chemical Sciences Laboratory (CSL), Boulder, CO 80305-3328, USA
[3]NASA Goddard Space Flight Center, Greenbelt, MD, USA
[4]Science Systems and Applications, Inc., Lanham, MD, USA
[5]TS1 Laboratory for Air Pollution/Environmental Technology, Empa, Swiss Federal Laboratories for Materials Science and Technologies, Duebendorf, Switzerland

**Correspondence:** Megan Jeramaz Lickley (mlickley@mit.edu)

**Abstract.** TS2 Halocarbons contained in equipment such as air conditioners, fire extinguishers, and foams continue to be emitted after production has ceased. These "banks" within equipment and applications are thus potential sources of future emissions, and must be carefully accounted for in order to differentiate nascent and potentially illegal production from legal banked emissions. Here, we build on a probabilistic Bayesian model, previously developed to quantify chlorofluorocarbon (CFC-11, CFC-12, and CFC-113) CE1 banks and their emissions. We extend this model to a suite of banked chemicals regulated under the Montreal Protocol (hydrochlorofluorocarbon, HCFC-22, HCFC-141b, and HCFC-142b, halon 1211 and halon 1301, and CFC-114 and CFC-115) along with CFC-11, CFC-12, and CFC-113 in order to quantify a fuller range of ozone-depleting substance (ODS) banks by chemical and equipment type. We show that if atmospheric lifetime and prior assumptions are accurate, banks are most likely larger than previous international assessments suggest, and that total production has probably been higher than reported. We identify that banks of greatest climate-relevance, as determined by global warming potential weighting, are largely concentrated in CFC-11 foams and CFC-12 and HCFC-22 non-hermetic refrigeration. Halons, CFC-11, and CFC-12 banks dominate the banks weighted by ozone depletion potential (ODP). Thus, we identify and quantify the uncertainties in substantial banks whose future emissions will contribute to future global warming and delay ozone-hole recovery if left unrecovered.

## 1 Introduction

The Montreal Protocol regulates the production of ozone-depleting substances (ODS), and its implementation has avoided a world with catastrophic stratospheric ozone depletion (Newman et al., 2009). Globally, there has been a near-cessation of chlorofluorocarbon (CFC) and halon production since 2010, and global production of the replacement hydrochlorofluorocarbons (HCFCs) is scheduled to be phased out by 2030. Despite production phase-out, these chemicals persist in old equipment produced prior to phase-out, such as refrigeration, air conditioners, foams, and fire extinguishers. These reservoirs of materials (termed "banks") continue to be sources of emissions (e.g., Carpenter et al., 2018). Previously published estimates of bank sizes and bank emissions vary widely due to different estimation techniques that incorpo-

rate incomplete or imprecise information (Kuijpers and Verdonik, 2009; Montzka et al., 2003). This uncertainty obscures the ongoing attribution of emissions and undermines international efforts to evaluate global compliance with the Montreal Protocol. In earlier work, Lickley et al. (2020, 2021) developed a Bayesian probabilistic banks model for CFCs that incorporates the widest range of constraints to date (Lickley et al., 2020, 2021). Here, we extend this model to the suite of major chemicals regulated by the Montreal Protocol that are subject to banking.

Previously published assessments typically rely on one of three modeling approaches to estimate bank sizes and then estimate emissions associated with these banks. In the "top-down" approach (e.g., Montzka et al., 2003), banks are estimated as the cumulative difference between reported production and observationally derived emissions. However, by taking the cumulative sum of a small difference between two large values, small biases in emissions or reported production estimates can propagate into large biases in bank estimates (Velders and Daniel, 2014). Some type of bias is thus expected since total production has very likely been greater than reported production due to both the under-reporting of production (e.g., Gamlen et al., 1986; Montzka et al., 2018) and the exclusion of point-of-production losses in reported production values. Further,estimates of emissions rely on observed concentrations along with global lifetime estimates, which have large uncertainties associated with them (Ko et al., 2013).

The second approach relies on a "bottom-up" accounting method (Ashford et al., 2004; Campbell et al., 2005) where the inventory of sales by equipment type are carefully tallied along with estimated release rates by application use. The bottom-up approach also relies on sales data from surveys of various equipment types and products as well as estimates of their respective leakage rates (Campbell et al., 2005). These are all subject to uncertainties, which contribute to uncertainties in bottom-up bank estimates as well. A limitation of the bottom-up accounting method is that observed atmospheric concentrations are used only as a qualitative check and are not explicitly accounted for in the analysis. Another important limitation is that data used in this method are unobserved and rather rely on estimated processes along with reported data, such as production or sales of equipment. Thus any bias in reporting could propagate into large biases in bank estimates.

The third approach, and the one used in more recent ozone assessments such as the World Meteorological Organization (WMO, 2011, 2018, 2014), uses a hybrid approach to calculate banks. Bottom-up banks estimated for 2008 are used as a starting point for the calculations. These banks are taken from Campbell et al. (2005) and represent interpolated values from the 2002 and 2015 estimates. The banks are then brought forward to the present time by adding the cumulate reported production and subtracting the cumulative observationally derived emission from 2008 through the present.

This approach is consistent with 2008 bottom-up bank estimates by design, however, as time between 2008 and the present has grown, the cumulative errors associated with the top-down approach become larger.

The modeling approach applied in the present study relies on Bayesian inference of banks (Lickley et al., 2020, 2021) where banks are estimated using an approach called the Bayesian Parameter Estimation (BPE). In this approach, a simulation model of the bottom-up method is developed, where prior distributions of input parameters are constructed from previously published values, accounting for large uncertainties in production and bank release rates. The simulation model simultaneously models banks, emissions, and atmospheric concentrations. Parameters in the simulation model are then conditioned (or updated) on observed concentrations by applying Bayes' Rule **CE2**. The final result is a posterior distribution of banks by chemical and equipment type, along with an updated estimate of production and release rates for each equipment type. This approach incorporates data and assumptions from both the bottom-up and top-down approaches, providing a simulation model consistent with the bottom-up accounting method while also being consistent with observed concentrations within their uncertainties.

The remainder of the paper includes the following: Sect. 2 presents the Bayesian modeling approach along with data used in the analysis. Section 3 provides a summary of the results of our analysis for each of the chemicals considered here. Finally, Sect. 4 provides a discussion of our primary findings and limitations of the analysis.

## 2   Methods

The Bayesian modeling approach from Lickley et al. (2020, 2021) draws on a Bayesian analysis approach called Bayesian melding, designed by Poole and Raftery (2000), that allows us to apply inference to a deterministic simulation model. We employ a version of this method that we henceforth refer to as the Bayesian Parameter Estimation (BPE), which allows for input parameter uncertainty (Hong et al., 2005; Bates et al., 2003). The model flow is implemented as follows: first we develop a deterministic simulation model, representing the "bottom-up" accounting method that simultaneously simulates banks, emissions, and mole fractions for each chemical and equipment type. In this analysis, the chemicals considered include CFC-11, CFC-12, CFC-113, CFC-114, CFC-115, HCFC-22, HCFC-141b, HCFC-142b, halon 1201, and halon 1311. Prior distributions for each of the input parameters are based on previously published estimates. We then specify the likelihood function as a function of the difference between observed and simulated mole fractions. Finally, we estimate posterior distributions of both the input and output parameters by implementing Bayes' Rule

using a sampling procedure. Each of the steps of the BPE are described in more detail below.

## 2.1 Simulation model

The simulation model, comprised of Eqs. (1)–(5), simultaneously models banks, emissions, and mole fractions for each chemical by equipment type for all years with available data up until 2019. Starting dates differ by chemical, see the Supplement TS4 for details. The simulation model is specified as follows:

$$B_{j,t+1} = (1 - RF_{j,t}) \times B_{j,t} + (1 - DE_{j,t}) \times P_{j,t}, \quad (1)$$

where $B_{j,t}$, is banks and $P_{j,t}$ is production of equipment category $j$ in year $t$. The fraction of the released bank is reflected by $RF_{j,t}$ and $DE_{j,t}$ reflects the fraction of production that is directly emitted in equipment category $j$ in year $t$. These same parameters are used to simulate emissions, $E_{j,t}$:

$$E_{j,t+1} = RF_{j,t} \times B_{j,t} + DE_{j,t} \times P_{j,t}. \quad (2)$$

Total banks, $B_{\text{Total},t}$, and total emissions, $E_{\text{Total},t}$, are then estimated as the sum across all $N$ equipment categories:

$$B_{\text{Total},t} = \sum_{j=1}^{N} B_{j,t}, \quad (3)$$

$$E_{\text{Total},t} = \sum_{j=1}^{N} E_{j,t}. \quad (4)$$

For chemicals where feedstock usage is reported, an additional term in Eq. (4) is included that accounts for feedstock emissions. Emissions, along with an assumed atmospheric lifetime, $\tau_t$, taken as the SPARC (2013) TS5 multimodel time-varying mean, are then used to simulate atmospheric mole fractions, $MF_t$:

$$MF_{t+1} = \exp\left(\frac{-1}{\tau_t}\right) \times MF_t + A \times E_{\text{Total},t}, \quad (5)$$

where $A$ is a constant that converts units of emissions by mass to units of mole fractions, and also takes into account a fixed factor of 1.07 taken from Daniel et al. (2007) TS6 that accounts for the discrepancy between surface mole fraction concentrations and the global mean value.

## 2.2 Prior distributions

The input parameters in the simulation model described above require initial values to be assigned, along with their probability distributions. These prior distributions ("priors") are developed to estimate mole fractions, emissions, and banks for CFC-11, CFC-12, CFC-113, CFC-114, CFC-115, HCFC-22, HCFC-141b, HCFC-142b, halon 1201, and halon 1311. Categories of bank equipment are defined by the categorization provided by the Alternative Fluorocarbons Environmental Acceptability Study (AFEAS 2001) which

varies by compound (shown in Table 1). For halons, there is a single category of bank (fire extinguishing agent).

The AFEAS data report global annual production up to 2001, categorized by equipment type, which is generally grouped as short, medium and long CE3 banks. We use AFEAS data and categorization to develop our production priors and adopt the WMO (2003) correction where AFEAS production values are used up until 1989 and then scaled to match the United Nations Environmental Programme's (UNEP) global production values for all years following 1989. After AFEAS data ends, we assume that the relative production in each category remains constant for all years following 2001. Uncertainty in production priors is assumed to follow a multivariate log-normal distribution, where temporal correlation in production reporting bias is estimated in the BPE. Prior distributions CE4 differ by chemical and are developed to be wide enough for atmospheric mole fraction priors to contain observations. See the Supplement TS7 for details on production priors for each chemical.

The emissions function by bank equipment type can be characterized by the fraction of production that is directly emitted during the year of production (DE) and the fraction of the bank that is emitted in each subsequent year. Prior estimates for the emissions function CE5 come from previously reported data and differ by chemical and equipment type (see the Supplement TS8). Broadly speaking, it has been estimated that chemicals contained in short banks are fully emitted within the first 2 years after production, medium banks lose about 10 %–20 % of their material each year, and long banks can lose as little as 2 % of their material each year (Ashford et al., 2004). We use previously published estimates to develop emissions function priors specific to each chemical and bank type along with wide uncertainties, as specified in the Supplement TS9.

Amounts of halocarbons used for feedstock production are available annually (UNEP/TEAP, 2021). A prior mean leakage rate of 2 % was assumed during production, which reflects an approximate average of values across different facilities (MCTOC, 2019).

## 2.3 Likelihood function

For each chemical, the likelihood function is a multivariate normal likelihood function of the difference between modeled and observed mole fractions:

$$P(D_{t1},\ldots D_{tN}|\theta) = \frac{1}{(2\pi)^{\frac{N}{2}}\sqrt{|S|}} \exp\left\{-\frac{1}{2}\Delta^T S^{-1}\Delta\right\}, \quad (6)$$

where $D_{t1},\ldots D_{tN}$ is yearly globally averaged observed mole fractions for all years where observations are available and TS10 $\theta$ represents that vector of input and output parameters from the simulation model. The $\Delta$ denotes an $N \times 1$ TS11 vector of the difference between yearly observed and modeled mole fractions and is assumed to have a mean zero, and covariance function $S$. Therefore, $S$ represents the sum of

**Table 1.** Application type of halocarbon banks by chemical.

| Chemical | Short bank | Medium bank | Long bank |
|---|---|---|---|
| CFC-11 | Aerosols Open-cell foam | Non-hermetic refrigeration | Closed-cell foam |
| CFC-12 | Aerosols Open-cell foam | Non-hermetic refrigeration | Refrigeration |
| CFC-113 | Solvents | | Heat pump |
| CFC-114 | | | Heat pump |
| CFC-115 | Propellant | | Air conditioning |
| HCFC-22 | Open-cell foam | Non-hermetic refrigeration | Foam |
| HCFC-141b | Open-cell foam | Non-hermetic refrigeration | Foam |
| HCFC-142b | | Non-hermetic refrigeration | Foam |
| Halon-1211 | | Fire extinguishing agent | |
| Halon-1301 | | Fire extinguishing agent | |

uncertainties between observed and modeled mole fractions. While there are published estimates of uncertainties in observed mole fractions, we do not know the uncertainties in modeled mole fractions. We therefore estimate $S$ separately for each chemical, as is done in Lickley et al. (2020). The off-diagonals in the covariance function incorporate a correlation term, $\rho_S$, which accounts for our assumption that there is high autocorrelation in the bias between modeled and observed mole fractions. Correlation terms for each chemical are reported in the Supplement TS12 along with prior estimates of the uncertainty parameters used for diagonal elements in $S$. Each column and row in $S$ is therefore populated as

$$S_{i,\,j} = \sigma_i \sigma_j \rho_S^{|i-j|},$$

where $\sigma_i$ and $\sigma_j$ represent the sum of the uncertainties in observed and modeled mole fractions at time $i$ and $j$, respectively, and are inferred in the BPE, whereas $\rho_S$ is prescribed.

Observations come from the Advanced Global Atmospheric Gas Experiment (AGAGE; https://agage.mit.edu, last access: TS13) data set (Prinn et al., 2000, 2018), with the exception of CFC-11 and CFC-12 which, following Lickley et al. (2021), come from the AGAGE and the National Oceanographic and Atmospheric Administration's (NOAA) merged data sets (Engel et al., 2019). Data are aggregated into annual global mean mole fractions. The time frame of availability of observations differs by chemical (see the Supplement TS14).

## 2.4 Posterior distributions

Following Bayes' Rule, we specify our posterior distribution as

$$P(\boldsymbol{\theta} \mid D_{t1}, \ldots, D_{tN}) = \frac{P(\boldsymbol{\theta}) \, P(D_{t1}, \ldots D_{tN} | \boldsymbol{\theta})}{P(D_{t1}, \ldots D_{tN})}, \qquad (7)$$

where $P(\boldsymbol{\theta})$ represents the joint prior distribution of the input and output parameters described in the simulation model in Sect. 2.1.

The analytical form of the posterior distribution is intractable. Thus, we estimate the posterior distribution CE6 using a sampling procedure (the sampling importance resampling (SIR) method) to estimate the marginal posterior distributions (Hong et al., 2005; Bates et al., 2003; Rubin, 1988). To implement SIR we draw 1 000 000 samples from the priors, run the simulation model, and then resample from the priors 100 000 times using an importance ratio, which is proportional to the likelihood function. These sample sizes were chosen such that multiple iterations of the model produce consistent results.

## 3 Results

Figure 1 shows observed globally averaged mole fractions compared to the BPE of CE7 mole fractions for each chemical. Figure 2 shows BPE and observationally derived emissions, assuming the SPARC multimodel time-varying mean lifetime for each species. Posterior estimates agree well with observations for the majority of time periods and chemicals. Note, however, that BPEs from Lickley et al. (2021) match observed and observationally derived estimates more closely for CFC-11 than they do in the present analysis. We attribute this difference in consistency to atmospheric lifetimes being assumed in the present analysis, while they were inferred in Lickley et al. (2021), who found inferred lifetimes to be somewhat shorter than the SPARC multimodel mean values CE8. Shorter lifetimes would allow modeled mole fractions to decline more quickly following 1990, matching observations better. A notable discrepancy occurs for CFC-115,

where modeled mole fractions are increasing throughout the entire simulation period, whereas observed mole fractions from 2000 onwards are relatively constant. This discrepancy could be explained by the large uncertainties in atmospheric lifetimes of CFC-115 (Vollmer et al., 2018), if atmospheric lifetimes are in fact substantially shorter than the SPARC multimodel mean.

Figure 3 provides a comparison of BPE bank estimates alongside previously published bank estimates. The BPE bank estimates are generally higher than other published values. This can be explained by production uncertainties that are accounted for in the present analysis. Our analysis suggests that production has most likely been underreported for nearly all chemicals. Table 2 provides a summary of our estimated bias in cumulative reported production throughout the simulation period for each chemical type. With the exception of CFC-113 and CFC-115, we find our inferred cumulative production to be significantly higher than reported production (at the 1-sigma level), with our median estimate suggesting that production was as little as 9 % higher than reported for CFC-12 and as high as 50 % higher than reported for halon 1211. Note, however, that high uncertainties in lifetimes for halon 1211 exist (Ko et al., 2013) and could explain part of this discrepancy. We would expect any consistent bias in reported production to be a bias low, since consistent undercounting of production is more plausible than overcounting production. The exception for this would be the base year, which refernces reduction targets CE10. In this instance, we would expect overreporting for this year to be more likely. Another possible explanation for the discrepancy in production estimates is that total reported chemical production under UNEP does not account for leakage during chemical manufacturing, but rather only leakage that occurs during the application of the chemical. To our knowledge, this potential leakage during chemical manufacturing has not been well-documented or previously quantified.

Figure 4 shows the breakdown of emissions by equipment type over time. For CFCs, emissions from short banks tend to peak around 1990, as spray applications were banned earlier than other applications, after which emissions from medium and long banks become more dominant emission sources. This is to be expected as the phase-out of production after 1990 would lead to more CFC emissions from existing banks rather than new, short-lived equipment. For HCFC-22, most of the emission throughout the entire time period is from medium banks, which is largely non-hermetic refrigeration. Long banks (i.e., foams) dominate emissions for HCFC-141b and for HCFC-142b, where both foams and non-hermetic refrigeration are prominent emission sources throughout the simulation period. Estimated feedstock emissions averaged over 2010–2019 are shown in Table 3. The HCFC-22 is the largest source of feedstock emissions by mass, but CFC-113 feedstock emissions are estimated to be larger when weighted by global warming potential (GWP100) and ODP.

Figure 5 shows the relative quantity of banked materials by chemical type. Banks are weighted by mass (Fig. 5a), by GWP100 (Fig. 5b), and ODP (Fig. 5c). Our best estimate is that the sum of the HCFCs currently comprise about 77 % of banks by mass. However, in terms of climate impacts, CFC-11, CFC-12, and HCFC-22 are the largest banked materials weighted by GWP100, accounting for 36 %, 14 %, and 36 % of current banks, respectively. When banks are weighted by ODP, CFC-11 and CFC-12 represent 46 % and halons also represent 46 % of current banked chemicals.

Figure 6 shows the composition of banks by chemical type. This, together with Fig. 5, provides insight into the most prominent banked sources of halocarbons with regards to GWP100 and ODP. In terms of GWP100, CFC-11 banks largely reside in foams, whereas CFC-12 and HCFC-22 are largely in non-hermetic refrigeration. The latter may be more readily recoverable. In terms of ODP, CFC-11 foams and CFC-12 non-hermetic refrigeration remain important, along with halons which are all contained in fire extinguishers, a recoverable reservoir.

## 4 Discussion and conclusions

This analysis suggests that if lifetime assumptions are correct, published bank estimates using either the top-down or bottom-up approaches were likely underestimating bank sizes for all banked chemicals due to underreporting of production (see Table 2). The Bayesian approach used in this analysis does not assume that production is known precisely, but rather jointly infers production along with the other parameters in the simulation model, providing probabilistic estimates of historical production values. Previously published bank estimates (Ashford et al., 2004; Kuijpers and Verdonik, 2009; Montzka et al., 2003) do not infer production, but rather assume that it is known, or consider different scenarios. We argue that production assumptions have been biased low due to underreporting of total production and potentially unaccounted for leakage during chemical manufacturing, and thus have led to published bank estimates that were also biased low.

Discrepancies between observed mole fractions and BPE-derived mole fractions are notable for the suite of chemicals considered here. While the majority fall within the 90 % confidence interval throughout most of the time periods, the trends in concentrations between observations and inferred mole fractions do not always agree. This discrepancy could be related to our partitioning of production type following 2003 (i.e., after AFEAS data end). Another important limitation in this analysis is in the treatment of atmospheric lifetimes, which could also explain some of these discrepancies. The present analysis assumes that atmospheric lifetimes are known and equal to the SPARC (2013) TS18 multimodel time-varying mean lifetimes. However, previous work has indicated potential biases in SPARC lifetimes, for example

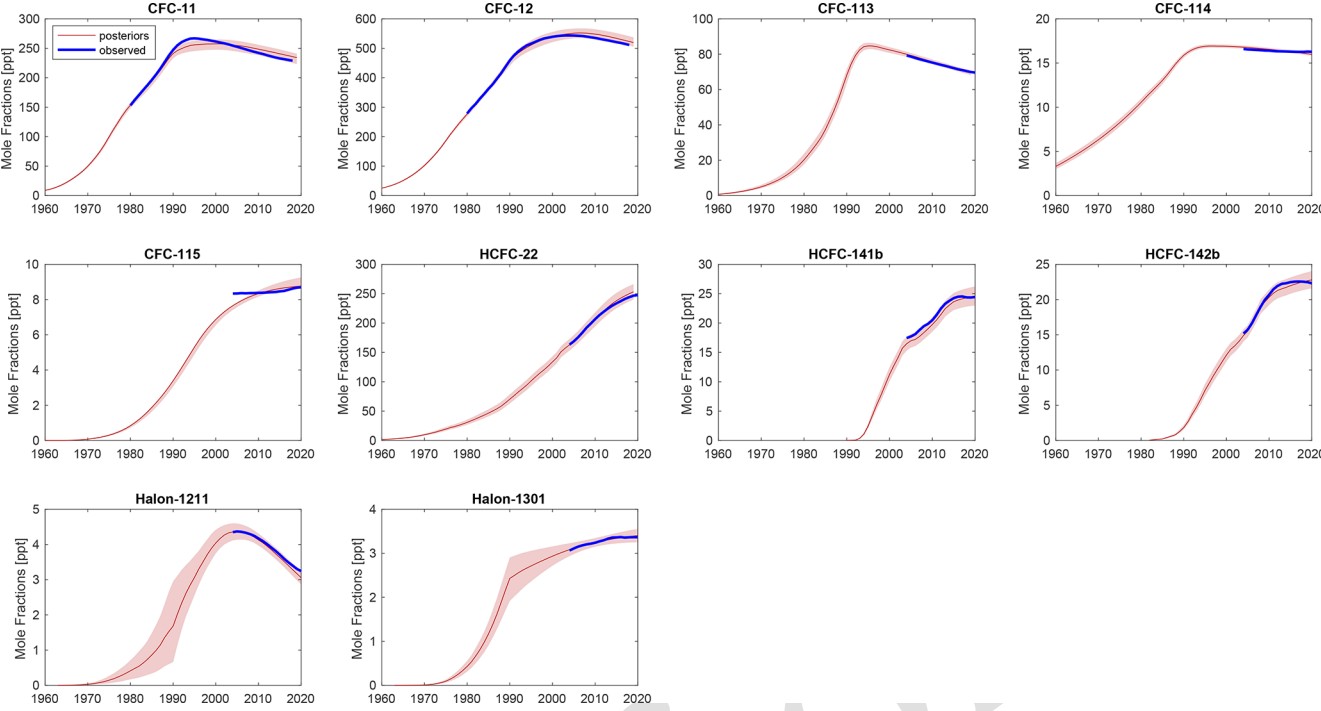

**Figure 1.** Modeled mole fractions versus observed mole fractions. Red lines indicate the posterior median mole fraction estimate from the Bayesian Parameter Estimation `CE9` (BPE), with shaded regions indicating the 90 % confidence interval. Blue lines indicate globally averaged observed mole fractions.

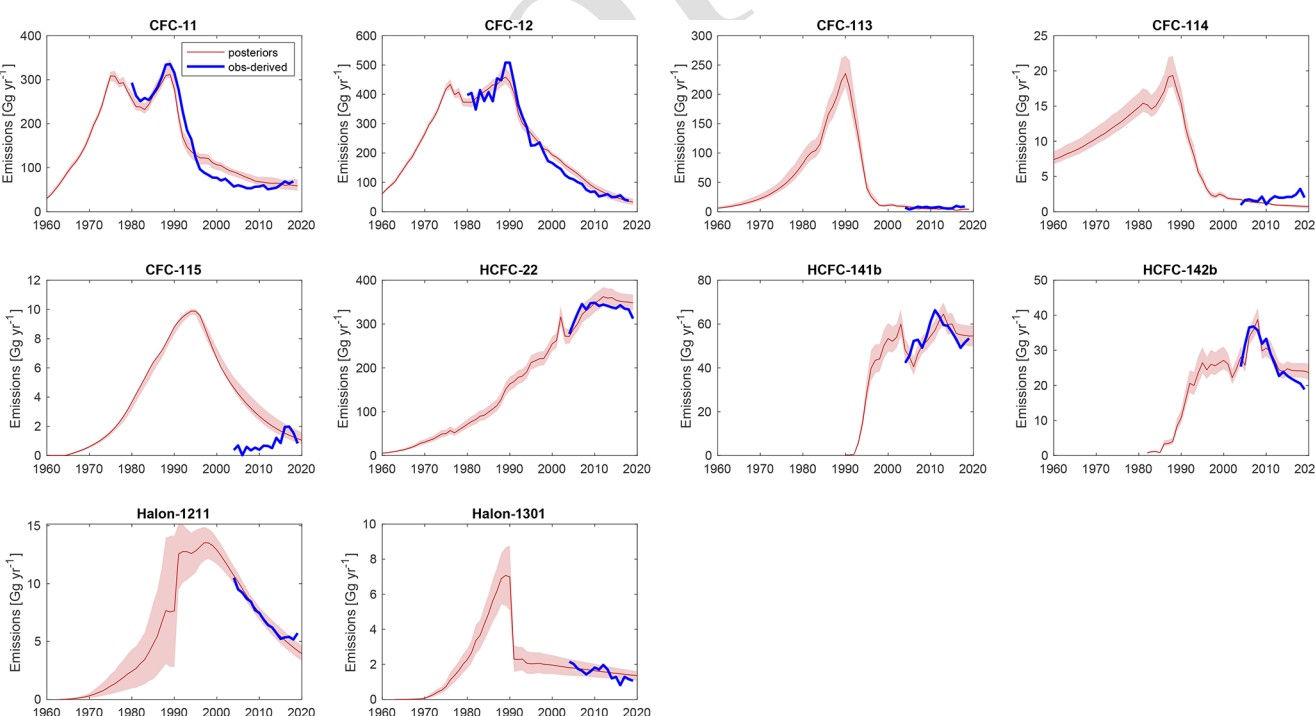

**Figure 2.** Modeled emissions versus observationally derived emissions. Red lines indicate the posterior median emissions estimate from the Bayesian Parameter Estimation (BPE), with shaded regions indicating the 90 % confidence interval. Blue lines indicate observationally derived emissions assuming the SPARC multimodel time-varying mean lifetimes.

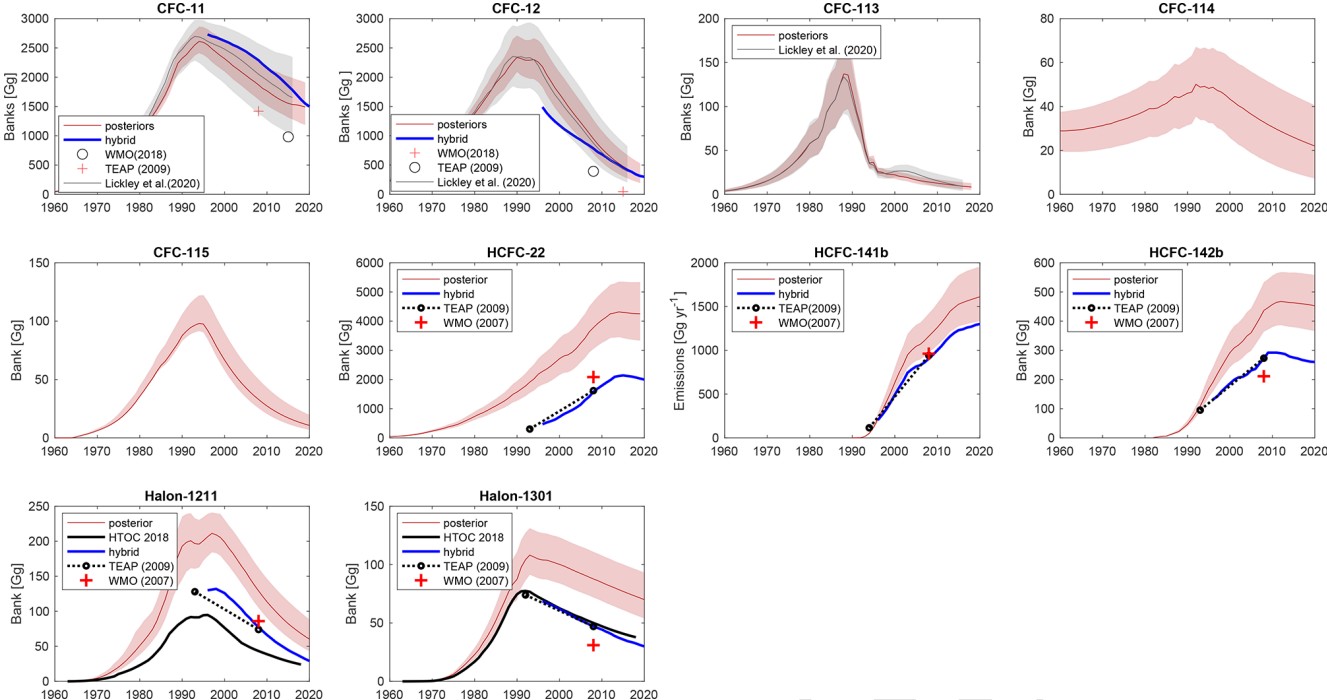

**Figure 3.** Magnitudes of bank estimates. The red lines indicate the median posterior estimate of banks from the Bayesian analysis, with shading indicating the 90 % confidence interval. Previously published bank estimates are provided for comparison from TEAP (2009) TS15, WMO (2007) TS16, WMO (2018) and Lickley et al. (2020) along with the hybrid approach updated to current estimated starting values.

**Table 2.** Estimated bias in cumulative reported production. Values indicate the percent difference between inferred cumulative production from the onset of production to 2019 relative to reported production, for all uses except feedstock production. Positive values indicate the percent by which inferred production is higher than reported.

| Chemical name | CFC-11 | CFC-12 | CFC-113 | CFC-114 | CFC-115 |
|---|---|---|---|---|---|
| Median percentage inferred bias (16th, 84th percentile) | 12 % (9 %, 13 %) | 9 % (7 %, 11 %) | −1 % (−3 %, 0 %) | 11 % (9 %, 13 %) | −1 % (−2 %, 5 %) |
| Median absolute inferred bias (16th, 84th percentile) [Gg] | 1146 (900, 1291) | 1208 (976, 1439) | −37 (−76, −3) | 58 (46, 70) | −2 (−4, 11) |
| Chemical name | HCFC-22 | HCFC-141b | HCFC-142b | Halon 1211 | Halon 1301 |
| Median percentage inferred bias (16th, 84th percentile) | 10 % (6 %, 13 %) | 12 % (6 %, 19 %) | 22 % (17 %, 28 %) | 50 % (41 %, 59 %) | 24 % (18 %, 32 %) |
| Median absolute inferred bias (16th, 84th percentile) [Gg] | 1249 (828, 1712) | 315 (153, 511) | 220 (166, 281) | 137 (114, 164) | 36 (26, 49) |

for CFCs (Lickley et al., 2021). The potential bias in atmospheric lifetimes would result in biased bank estimates in the present paper and requires further analysis.

This modeling approach makes no assumptions about end-of-life (EOL) emissions. Certain bank estimates assume that applications are dismantled at the end of their lifetime, which would contribute to both decreased banks and increased emissions at fixed years after production (e.g., UNEP/TEAP, 2019). We do not make this assumption as we believe it would be more realistic for dismantling of equipment to occur over a range of years after production, which would effectively be captured by our bank release fraction estimate.

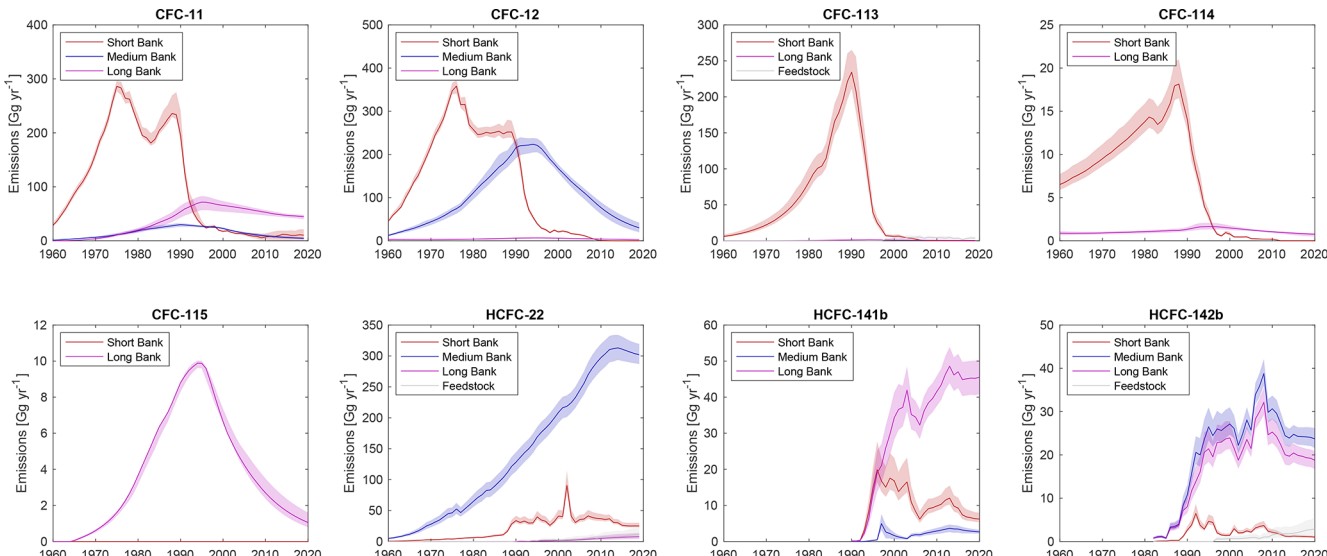

**Figure 4.** Emissions by source – Estimates of emissions by various equipment types, summarized in Table 1, are shown here along with estimated emissions from feedstock usage. Lines indicate the median estimate, with the shaded region indicating the 90 % confidence interval. Halons are not included in this figure as 100 % of halon emissions come from the same application and are thus identical to Fig. 2 halon totals.

**Table 3.** Estimated feedstock emissions averaged from 2010–2019 from the Bayesian analysis. Emissions are weighted by mass, global warming potential (GWP100) relative to $CO_2$ over a 100-year time horizon for a $CO_2$ concentration of 391 ppm, and by ozone depletion potential (ODP) relative to CFC-11 (WMO, 2018).

| Feedstockemissions | CFC-113 | HCFC-22 | HCFC-142b |
|---|---|---|---|
| By mass | 3.4 Gg yr$^{-1}$ | 9.3 Gg yr$^{-1}$ | 2.1 Gg yr$^{-1}$ |
| By GWP100 | 20 838 Gg yr$^{-1}$ | 16 591 Gg yr$^{-1}$ | 4302 Gg yr$^{-1}$ |
| By ODP | 2.8 Gg yr$^{-1}$ | 0.3 Gg yr$^{-1}$ | 0.1 Gg yr$^{-1}$ |

We do however test the sensitivity of our bank estimate to EOL emissions occurring in a single year after production. This we term the EOL scenario and test the sensitivity of banks for CFC-11, CFC-12, and HCFC-22, the three largest banks by global warming potential. The modeling approach is described in the Supplement CE11 and results are shown in Fig. SM1 therein. Perhaps unexpectedly, posterior bank estimates of CFC-11 are $\sim 25\%$ higher in 2020 in the EOL scenario relative to the scenario described in the main text. However, banks in the EOL scenario are decreasing faster than those described in the main text. The larger bank size is due to posterior bank release fractions being $\sim 2\%$ for the EOL scenario relative to 3 % for the scenario described in the main text. The faster depletion of the banks in 2020 can be explained by the addition of the EOL decommissioning parameter. These larger bank estimates reflect the consistency of the Bayesian modeling approach where all parameters are jointly inferred. Including an additional process in the model requires that multiple parameters be updated to be consistent with observations. For CFC-12, the EOL scenario produces significantly smaller banks from about 1990 onwards. How-

ever, the emissions profile has an artificial dip in emissions relative to observationally derived emissions, suggesting that a set year for decommissioning is not a realistic modeling assumption. For HCFC-22, banks are not substantially different between the two scenarios.

There are important discrepancies between CFC-113 feedstock emissions inferred here and those estimated in the previous analysis (Lickley et al., 2020). In Lickley et al. (2020), feedstock emissions were assumed to be the difference between observationally derived emissions and inferred bank emissions. In the present analysis, prior distributions CE12 of feedstock production and leakage rates are developed and feedstock emissions are then inferred. In the present analysis, observationally derived CFC-113 emissions are higher than total BPE-inferred emissions at the 1-sigma level from 2010 onwards. This suggests that either observationally derived emissions are too high, or our BPEs are too low. In Lickley et al. (2021), we find that atmospheric lifetimes of CFC-113 are most likely lower than the SPARC multimodel time-varying mean used in the present analysis. This would imply that the observationally derived emissions shown in

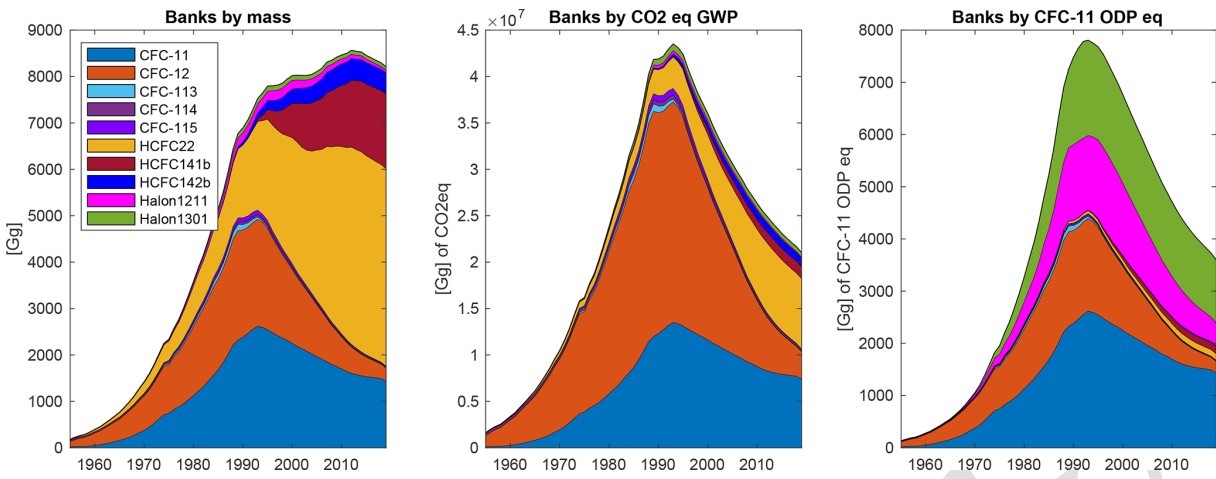

**Figure 5.** Total banks by mass, global warming potential (GWP100; WMO, 2018), and ozone depleting potential (ODP; WMO, 2018). Bank estimates reported in the above figures are the median estimates from the Bayesian analysis.

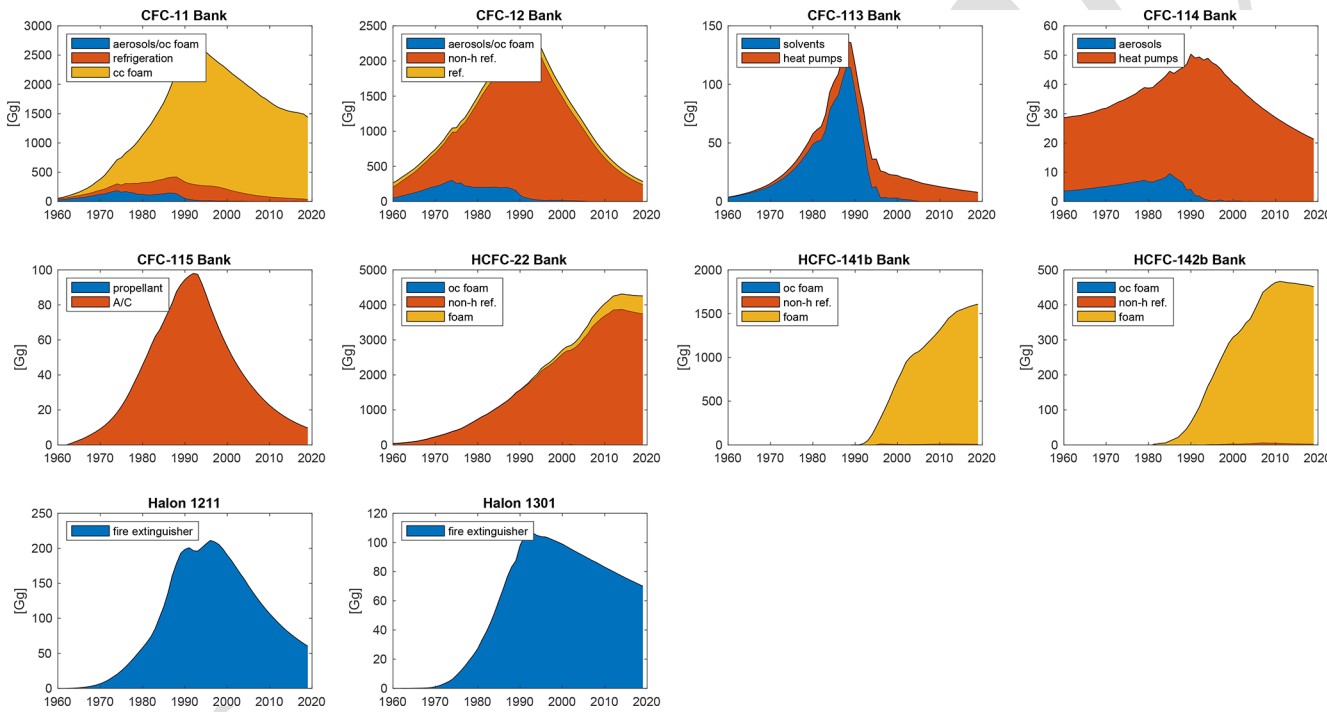

**Figure 6.** Bank size by equipment type. Bank estimates reported in the above figures are the median estimates from the Bayesian analysis. In the above legends, "cc" refers to closed-cell foams, "non-h ref." refers to non-hermetic refrigeration, "ref." refers to refrigeration, and "A/C" refers to air conditioning.

Fig. 2 are biased low, suggesting an even larger discrepancy between BPE-inferred total emissions and observationally derived emissions. Therefore, it seems plausible that the discrepancy is due to prior feedstock emissions estimates being biased low due to larger leakage, or CFC-113 is being produced for a use that is not allowed under the Montreal Protocol.

Finally, some important details about production and destruction were not fully accounted for in this analysis. For one, feedstock priors were only included for CFC-113, HCFC-22, and HCFC-142b, which could be limiting our assessment of the sources of emissions for other chemicals. However, published feedstock values for other chemicals are not available and leakage rates in feedstock applications may be uncertain. In addition, we do not account for non-dispersive production in our analysis, namely the production of chemicals as by-products. It is possible, for example, that some of the discrepancies in CFC-115 emissions could be ex-

plained by non-dispersive emissions as identified by Vollmer et al. (2018). Moreover, we do not consider EOL destruction of equipment as there are no published records, to our knowledge, of these processes. Finally, we were not able to account for a more detailed breakdown in production by equipment type than what has been published by AFEAS, which discretizes production into, at most, four categories of equipment, and does not provide data beyond 2003. Without publicly available details of these processes, modeling of banks and emissions will continue to be limited.

**Code availability.** All analyses were done in MATLAB. All code used in this work is available at https://github.com/meglickley/HalocarbonBanks [TS19].

**Data availability.** The data sets generated and/or analyzed during the current study are available at https://github.com/meglickley/HalocarbonBanks [TS20].

**Supplement.** The supplement related to this article is available online at: https://doi.org/10.5194/acp-22-1-2022-supplement.

**Author contributions.** All authors contributed to the conceptualization of the manuscript. MJL conducted the analysis. MJL prepared the manuscript with contributions from all authors. [TS21]

**Competing interests.** The contact author has declared that none of the authors has any competing interests.

**Disclaimer.** Publisher's note: Copernicus Publications remains neutral with regard to jurisdictional claims in published maps and institutional affiliations.

**Acknowledgements.** Megan Jeramaz Lickley and Stefan Reimann gratefully acknowledge the support of VoLo foundation and grant 2128617 from the Atmospheric Chemistry Division of the National Science Foundation. AGAGE is principally supported by NASA (USA) grants to MIT and SIO, and also by BEIS (UK) and NOAA (USA) grants to Bristol University; CSIRO and BoM (Australia): FOEN grants to Empa (Switzerland); NILU (Norway); SNU (Korea); CMA (China); NIES (Japan); and Urbino University (Italy). Eric Fleming acknowledges support of the NASA Headquarters' Atmospheric Composition Modeling and Analysis Program (ACMAP).

**Financial support.** . [TS22]

**Review statement.** This paper was edited by Farahnaz Khosrawi and reviewed by three anonymous referees.

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

## Remarks from the language copy-editor

CE1 Please confirm the change.

CE2 Please note that "theorem" has been changed to "Rule".

CE3 Please note that "long-term" has been changed to "long" as per Table 1. Please confirm.

CE4 Do you mean "Priors"?

CE5 Please confirm the change.

CE6 Please confirm the change.

CE7 Please note that "estimated" has been removed because the abbreviation already contains "estimation".

CE8 Please check that the meaning of your sentence is intact.

CE9 Please note that "analysis" has been changed to "Parameter Estimation".

CE10 Please check that the meaning of your sentence is intact.

CE11 Please note that "SM" has been changed to "Supplement".

CE12 Do you mean "priors"?

## Remarks from the typesetter

TS1 According to our house standards, units in the affiliations should be listed from smallest to largest. Please check that this is the case here.

TS2 Please send a new Supplement as a *.pdf without the title, authors, correspondence author, etc. as we will generate a Supplement title page during publication (with a citation including the DOI), which will contain this information.

TS3 Please confirm running title.

TS4 Would you like to specify which part of the Supplement shows the corresponding information?

TS5 This reference is not in the reference list. Please check.

TS6 This reference is not in the reference list. Please check.

TS7 Would you like to specify which part of the Supplement shows the corresponding information?

TS8 Would you like to specify which part of the Supplement shows the corresponding information?

TS9 Would you like to specify which part of the Supplement shows the corresponding information?

TS10 Please check throughout the text that all vectors are denoted by bold italics and matrices by bold roman.

TS11 Please confirm change of "x" to "×".

TS12 Would you like to specify which part of the Supplement shows the corresponding information?

TS13 Please provide date of last access.

TS14 Would you like to specify which part of the Supplement shows the corresponding information?

TS15 Do you mean "Kuijpers and Verdonik, 2009"? "TEAP (2009)" is not in the reference list.

TS16 Not in reference list. Please check.

TS17 Please note that units have been changed to exponential format throughout the text. Please check all instances.

TS18 Not in reference list. Please check.

TS19 Please clarify whether the data set is your own. If yes, please provide a DOI in addition to your GitHub URL since our reference standard includes DOIs rather than URLs. If you have not yet created a DOI for your data set, please issue a Zenodo DOI (https://help.github.com/en/github/creating-cloning-and-archiving-repositories/referencing-and-citing-content). If the data set is not your own, please inform us accordingly. In any case, please ensure that you include a reference list entry corresponding to the data set including creators, title, and date of last access.

TS20 Please clarify whether the data set is your own. If yes, please provide a DOI in addition to your GitHub URL since our reference standard includes DOIs rather than URLs. If you have not yet created a DOI for your data set, please issue a Zenodo DOI (https://help.github.com/en/github/creating-cloning-and-archiving-repositories/referencing-and-citing-content). If the data set is not your own, please inform us accordingly. In any case, please ensure that you include a reference list entry corresponding to the data set including creators, title, and date of last access.

TS21 It should be clear who contributed to which part of the manuscript. For example, there are guidelines (see https://publications.copernicus.org/for_authors/obligations_for_authors.html) for who may be listed as a co-author. Please provide us with a more specific text (complete sentences) for the Author contributions section.

TS22 Please note that there is funding information given in the acknowledgements, but you did not indicate any funding upon manuscript registration. Therefore, we were not able to complete the financial support statement. Please provide the missing information and double-check your acknowledgements to see whether repeated information can be removed from the acknowledgements. Thanks.

TS23 Please ensure that any data sets and software codes used in this work are properly cited in the text and included in this reference list. Thereby, please keep our reference style in mind, including creators, titles, publisher/repository, persistent identifier, and publication year. Regarding the publisher/repository, please add "[data set]" or "[code]" to the entry (e.g. Zenodo [code]).

TS24 Please provide date of last access.

TS25 Please check journal name.

TS26 Please provide edition (if any), page numbers, and DOI or URL including last access date.

TS27 Please provide edition (if any), editors (if any) and DOI or URL including last access date.

TS28 Please provide edition (if any), editors (if any) and DOI or URL including last access date.

TS29 Please provide page range or article number and DOI.

TS30 Please provide edition (if any), publisher, page range, an DOI or URL including last access date.

TS31 Please provide more information (e.g. a DOI or URL including last access date).

TS32 Please provide page range or article number.

TS33 Please provide more information (e.g. a DOI or URL including last access date).

TS34 Please provide edition (if any), editors (if any), publisher, page numbers, and ISBN or DOI.

TS35 Please provide DOI.

TS36 Please provide more information (e.g. a DOI or URL including last access date).

TS37 Please provide more information (e.g. a DOI or URL including last access date).

TS38 Please provide DOI or URL including last access date.

TS39 Please provide DOI or URL including last access date.

TS40 Please provide DOI or URL including last access date.

TS41 Please provide DOI or URL including last access date.