# Peer review of "Bayesian assessment of chlorofluorocarbon (CFC), hydrochlorofluorocarbon (HCFC) and halon banks suggest large reservoirs still present in old equipment"

_Atmospheric Chemistry and Physics, 2022_

## Author Response (AR1)

**Ref #1**

Review for *"Bayesian assessment of chlorofluorocarbon (CFC), hydrochlorofluorocarbon (HCFC) and 2 halon banks suggest large reservoirs still present in old equipment"* by Lickley et al.

This manuscript presents an extension to earlier work, where ODS banks, and their emissions, are estimated in a statistical framework using uncertain knowledge about emission profiles and atmospheric mole fraction measurements. This study is timely, clear and concise, and will be of broad interest to readers of ACP. I have only a few suggestions, with the majority concerning clarifications in the text and some additional discussion. I have one concern over the treatment of what is termed 'reported production' and how this may impact the conclusions drawn, which I outline below. I hope to see the eventual acceptance and publication of this manuscript once my comments have been addressed.

**Comments:**

1. **Production-related emissions and leakage rates:** From my understanding, reported production is the net production, i.e. reported after any losses during the production process, or the "sellable production". As such, at least in theory, cumulative production should equal consumption for CFCs. This means that total production, before any production losses, would be larger than that reported, i.e. Total production = P/(1-DE) using the notation in the paper (where P is reported production), and production-related emissions would be equal to DE*P/(1-DE). What I take from the manuscript is that, currently, the production-related emissions are quantified instead from the production that has been reported after the losses have occurred. I don't think this will alter the conclusions of the paper, but it may close some of the gap and reduce some of the reported bias, as it may mean a few percent more production is added to the bank.

The reviewer raises an important point that has not been well-documented in previous literature. In the present manuscript we have not accounted for leakage during chemical production, which may very well account for some of the differences between our estimated production and reported production/consumption. We have added this to the results and discussion of estimated production bias: Lines 307-312:

"Another possible explanation for the discrepancy in production estimates is that total reported chemical production under the UNEP does not account for leakage during chemical manufacturing, but rather only leakage that occurs during the application of the chemical. To our knowledge, this potential leakage during chemical manufacturing has not been well-documented or previously quantified."

Lines 371-372

"and potentially unaccounted for leakage during chemical manufacturing"

2. **Discussion of discrepancies in production:** Non-dispersive (and therefore not required to be reported) production exists, for example when gases are produced as by-products. An example CFC-115 contamination in the production of HFC-125 (see Vollmer et al 2018). I don't believe this is a reason for the discrepancies, and there's no evidence to support or deny this, but should be discussed in addition to dispersive and feedstock related production.

This is an excellent point. We have added this citation and discussion to our final paragraph in the manuscript, Lines 429-432:

"In addition, we do not account for non-dispersive production in our analysis, namely the production of chemicals as by-products. It is possible, for example, that some of the discrepancies in CFC-115 emissions could be explained by non-dispersive emissions as identified by (Vollmer et al., 2018)."

3. **End-of-life emissions:** The impact of end-of-life emissions needs further discussion. How would a change in the emissions rate due to disposal change the conclusions drawn? Some information exists surrounding post-disposal emissions, e.g. from the US EPA (see e.g. page A-262, Table A-127 of https://www.epa.gov/system/files/documents/2022-04/us-ghg-inventory-2022-annex-3-additional-source-or-sink-categories-part-a.pdf). I imagine this would be too difficult to include in any analysis (and I'm not sure if there is sufficient information to do this) but there is a need to expand the discussion around long-lived banks.

We've added an end-of-life emissions sensitivity analysis to the supplement and describe how bank estimates can be impacted by this additional process. In a bottom-up modeling approach, where are all parameters are assumed known, adding an end-of-life emissions would result in a smaller bank estimate. However, in the Bayesian modeling approach, all parameters are inferred and interdependent, so results are not as straight forward. We find that CFC-11 banks end up being higher if we assume end-of-life emissions are occurring and lower for CFC-12. We explain this in the discussion on Lines 387-408 and include results and updated modeling equations in the SM:

"This modeling approach makes no assumptions about end-of-life emissions. Certain bank estimates assume that applications are dismantled at the end of their lifetime, which would both contribute to decreased banks and increased emissions at fixed years after production (e.g. TEAP, 2019). We do not make this assumption as we believe it would be more realistic for dismantling of equipment to occur over a range of years after production, which would effectively be captured by our bank release fraction estimate. We do, however, test the sensitivity of our bank estimate to end-of-life (EOL) emissions occurring in a single year after production. This we term the EOL scenario and test the sensitivity of banks for CFC-11, CFC-12 and HCFC-22, the three largest banks by global warming potential. The modeling approach is described in the SM and results are shown in Figure SM1. Perhaps unexpectedly, CFC-11 posterior bank estimates are ~25% higher in 2020 in the EOL scenario relative to the scenario described in the main text. However, banks in the EOL scenario are decreasing faster than those described in the main text. The larger bank size is due to posterior bank release fractions being ~ 2% for the EOL scenario relative to 3% for the scenario described in the main text. The faster depletion of the banks in 2020 can be explained by the addition of the EOL decommissioning

parameter. These larger bank estimates reflect the consistency of the Bayesian modeling approach where all parameters are jointly inferred. Including an additional process in the model requires that multiple parameters be updated to be consistent with observations. For CFC-12, the EOL scenario produces significantly smaller banks from about 1990 onwards, however, the emissions profile has an artificial dip in emissions relative to observationally-derived emissions, suggesting a set year for decommissioning is not a realistic modeling assumption. For HCFC-22 banks are not substantially different between the two scenarios. "

**Technical comments:**

Abstract, line 19: "…must be carefully accounted for in order to evaluate ongoing compliance with the Montreal Protocol". This could be interpreted that these future emissions fall under controls of the Montreal Protocol, even though emissions from the bank are not controlled. Perhaps better is suggest the importance is to evaluate nascent production vs. banked emissions, or similarly you could change the 'impact' to be for something like stratospheric ozone recovery.

We made this change.

Abstract, line 22: "model a suite" rather than "the suite"

We made this change.

Abstract, line 27: "production for dispersive uses" or similar, as production will be higher than that reported due to e.g. by-product emissions and point of production losses.

We added the term "total production" here, but otherwise kept the wording. We don't think it is obvious to the scientific community that production should be higher than reported, as the reviewer has pointed out. Instead, we clarify this in the introduction, results and discussion. Lines 69-72:

"Some type of bias is thus expected since total production has very likely been greater than reported production both due to under-reporting of production (e.g. Gamlen et al., 1986; Montzka et al., 2018) and due to the exclusion of point of production losses in reported production values."

Lines 307 – 312:

"Another possible explanation for the discrepancy in production estimates is that total reported chemical production under the UNEP does not account for leakage during chemical manufacturing, but rather only leakage that occurs during the application of the chemical. To our knowledge, this potential leakage during chemical manufacturing has not been well-documented or previously quantified."

And Lines 370 -373:

"We argue that production assumptions have been biased low due to underreporting of total production and potentially unaccounted for leakage during chemical manufacturing and thus have led to published bank estimates that were also biased low."

Abstract, line 32: Delay ozone recovery in reference to what? Current projection generally take banks into account.

We think this is clear in the sentence that this is in reference to a future world without banks, because we end the sentence explaining "if left unrecovered".

Line 38: *stratospheric* ozone depletion

We made this correction.

Line 48: On the use of "we developed…". "we" here does not envelop all co-authors, so perhaps better to refer to it either in the passive, or "Lickley et al 2020, 2021 developed…"

Thank you, we made this correction.

Line 55-57: It would be clearer to continue to use the term 'reported production' rather than just 'production' here.

Thank you, we made this correction and added some clarification about total production versus reported production discrepancies.

Line 75: Are these cumulative emissions derived using a top-down or bottom-up method?

Top-down. We added clarifying text by calling them "observationally-derived emissions".

Line 86: It's generally called "Bayes' theorem" rather than "theory"

Thank you. We made this correction.

Line 127, eq. 3,4: Subscript italics should be saved for variables, i.e. 'Total' should not be italic.

Thank you. We made this correction.

Line 141, eq 5: It is not clear where the constants A come from. Is it from the SPARC 2013 reference? If not they should be included in the manuscript.

A differs by chemical as it depends on molecular weight. It also accounts for the difference between measured surface concentrations and global atmospheric means, following Daniel et al. (2007), Lines 175 - 177

"where A is a constant that converts units of emissions by mass to units of mole fractions, and also takes into account a fixed factor of 1.07 taken from Daniel et al. (2007) that accounts for the discrepancy between surface mole fraction concentrations to global mean value."

Line 149: Does the term 'fire extinguishers' here refer to all forms of fire and explosive suppressant? Halons are used in many other applications that only fire extinguishers. If this is only a general term used by AFEAS then this should be made clearer.

We've corrected this to be 'fire extinguishing agents', which would be more all-encompassing and reflects the wording of NOAA's Global Monitoring Laboratory. https://gml.noaa.gov/hats/about/halon.html

Line 157: It's not clear what functional form the correlation takes (this is also not clear in the supplement), only the prior set on the correlation parameter is specified. I.e. is there an assumed linear covariance? Or something like autocorrelation?

Yes – the correlation parameter accounts for autocorrelation and used to construct the off-diagonals in covariance matrix used in the likelihood function. We specified that the correlation parameter reflects the autocorrelation in the difference between modeled and observed MF. We also added some text and specified the elements of S explicitly on Lines 243 - 247:

"Each column and row in $S$ is therefore populated as;

$$S_{i,j} = \sigma_i \sigma_j \rho_S^{|i-j|}$$

where $\sigma_i$ and $\sigma_j$ represent the sum of the uncertainties in observed and modeled mole fractions at time $i$ and $j$, resepectively, and are inferred in the BPE, whereas $\rho_S$ is prescribed. "

Line 173: What's a 'medium value'? Should this be median?

This term was used to reflect an approximate average of the rate of feedstock emissions across facilities which is noted to be as high as 5% of production and as low as 0.5% of production in the MCTOC report. We changed the wording to be "approximate average".

Line 189: There should be a full stop instead of a comma before 'therefore'

We made this correction.

Line 196: NOAA needs defining

We added this correction.

Line 422: The reference to Vollmer et al. 2018 has a formatting error

This has been fixed.

Table 2: It would be very useful to also include the absolute difference in the estimates (with uncertainties) here in Gg, in additional to only the percentages, to get a better sense of the importance of these various compounds.

We added this change to the table.

Figure 3: Is there a significance to the circle size in Fig 3? If this is to distinguish between WMO 2018 and TEAP 2009 only then perhaps choose a different shape or line style.

We changed the circles to distinguish between TEAP and WMO.

**Ref #2**

This is a nice extension of a previous analysis to include more chemicals and more processes affecting past and potentially future emissions of halocarbons from banks. The work is important and highly relevant to current issues about halocarbons and Montreal Protocol compliance. I had 2 main issues that need some discussion and exploration before the paper is publishable.

Some consideration of end-of-life processes (sensitivity or includes as a separate category) is needed for banks for which end-of-life emission might be substantially different from emission rates during use (particularly cc foam for CFC-11, perhaps also non-hermetic refrigeration for CFC-12 and HCFC-22). TEAP reports have suggested that this could be a significant influence on emissions in recent and future years, yet this process is not considered by the authors owing to their view of a lack of information (line 329). I'd suggest that some exploration or sensitivity analysis of the issue is important to increase the relevance of these results. The Bayesian approach provides optimized parameters for the past, and those parameters may not be relevant for the future when the relative contributions of end-of-life emission increases substantially. For the model to provide useful expectations of emissions in the future, it must accurately represent a future where emissions are dominated by processes not as prominent in the past, i.e., end-of-life.

As described above and as suggested by the reviewer, we've added an end-of-life emissions sensitivity analysis to the supplement and describe how bank estimates can be impacted by this additional process. In a bottom-up modeling approach, where are all parameters are assumed known, adding an end-of-life emissions would result in a smaller bank estimate. However, in the Bayesian modeling approach, all parameters are inferred and interdependent, so results are not as straight forward. We find that CFC-11 banks end up being higher if we assume end-of-life emissions are occurring and lower for CFC-12. We explain this in the discussion on Lines 387 - 408 and include results and updated modeling equations in the SM:

"This modeling approach makes no assumptions about end-of-life emissions. Certain bank estimates assume that applications are dismantled at the end of their lifetime, which would both contribute to decreased banks and increased emissions at fixed years after production (e.g.TEAP, 2019). We do not make this assumption as we believe it would be more realistic for dismantling of equipment to occur over a range of years after production, which would effectively be captured by our bank release fraction estimate. We do, however, test the sensitivity of our bank estimate to end-of-life (EOL) emissions occurring in a single year after production. This we term the EOL scenario and test the sensitivity of banks for CFC-11, CFC-12 and HCFC-22, the three largest banks by global warming potential. The modeling approach is described in the SM and results are shown in Figure SM1. Perhaps unexpectedly, CFC-11 posterior bank estimates are ~25% higher in 2020 in the EOL scenario relative to the scenario described in the main text. However, banks in the EOL scenario are decreasing faster than those described in the main text. The larger bank size is due to posterior bank release fractions being ~ 2% for the EOL scenario

relative to 3% for the scenario described in the main text.  The faster depletion of the banks in 2020 can be explained by the addition of the EOL decommissioning parameter. These larger bank estimates reflect the consistency of the Bayesian modeling approach where all parameters are jointly inferred.  Including an additional process in the model requires that multiple parameters be updated to be consistent with observations.  For CFC-12, the EOL scenario produces significantly smaller banks from about 1990 onwards, however, the emissions profile has an artificial dip in emissions relative to observationally-derived emissions, suggesting a set year for decommissioning is not a realistic modeling assumption.  For HCFC-22 banks are not substantially different between the two scenarios. "

It isn't clear if the AGAGE mole fractions being fit in this analysis are surface means or some representation of total atmosphere average mole fractions.  I suspect that observation-based surface mean mole fractions are being used 'as is' to represent total atmosphere mean abundances and, if so, some further consideration is needed. For nearly all of these gases the bias between these two quantities was substantial in the past (up to 20% theoretically but likely less for most years), varies over time (reduced in recent years as emissions are less), and might add substantially to the larger than reported production and estimated bank sizes argued for in the present analysis. Related to this point (vertical mole fraction gradients are substantial and time-varying) a more realistic and time-varying relationship between mole fraction and emission (equation 5) needs including if indeed surface mole fractions are what is included in the inversion analysis.

This is an important point raised by the reviewer. We have used Global mean surface mole fractions from AGAGE, and account for the discrepancy between surface concentrations and global means by adjusting the data by a correction factor of 1.07 following Daniel et al. (2007). As Daniel et al. note, "The factor will certainly vary with trends in surface emissions; it is thus, in reality, a function of both time and compound and is a source of some uncertainty in the emission estimate."  In the present manuscript we do use time-varying lifetimes following the SPARC CCM lifetimes, but we've adopted a constant correction factor and have stated that our results are contingent on these assumptions.   We feel that narrowing the uncertainty of lifetimes and corrections factors are not within the scope of this paper.

Details:

The abstract makes assertions that seem too strong given the substantial caveats mentioned at the very end (lines 325-334). These caveats seem outside the assumptions related to priors and lifetime that are mentioned or even hinted at in the abstract.

Through the review process, it has been brought to our attention that reported production does not account for dispersive emissions.  Given this, we do think it is virtually certain that total chemical production has systematically exceeding reported production.  Therefore, we think the abstract reflects this.

Line 69, accounting methods use reported information, but also estimates and assumptions about processes leading to emissions.

We changed this to:

"rely on estimated processes along with reported data"

Figures 1 and 2 show observationally-based results and emissions for only a fraction of years for a number of gases. Inversions would seem to be less relevant and less accurate if performed on these limited data histories.

It's true that more data would provide more checks on our simulation model, however, insights on processes prior to observations are still gained through the inversion. The simulation model first simulates the entire time series of banks and emissions beginning the year when production begins, which is before observations are available for any of the chemicals considered here. For the posterior, the entire timeseries is then conditioned on available data, so that portions of the processes outside of the observational record are still updated through the inversion.

Citations of Assessment reports should be called out by the lead authors' names in most if not all places (not WMO 2018 or SPARC, 2013). It is only done the accepted way in a few instances in the manuscript.

Thank you. We made these changes throughout.

Line 321-3. Are these really the only two possible explanations for 113?

The reason for the emissions gap for 113 is not clear. The discrepancies that we note in the paper could encompass a range of activities as we note that 113 could be produced for a use that is not permitted under the Protocol. Banks of 113 are very likely too small to explain the emissions discrepancy (see for ex. Lickley et al. 2020).

**Ref #3**

This well-written manuscript uses a probabilistic Bayesian model to quantify residual storage (banks) of multiple ozone-depleting substances that are released to the atmosphere even after their production has been curtailed by regulation. The method appears to be a valuable approach to checking compliance to the Montreal Protocol for multiple compounds, and earlier iterations of this method have been used to infer lifetimes and banks of CFC-11, CFC-12 and CFC-113. From my reading, it appears to mesh fairly well with observation-based approaches that use background concentrations, global transport models, and inverse modeling to derive emissions estimates (e.g., the concurrently submitted paper to this journal on HCFC-142b by Western *et al*). I am not intimately familiar with either modeling approaches, so my goal here is to enhance the readability of the present manuscript to expand accessibility to larger audiences. The following are suggested as items for clarification.

1. Introducing the terms "prior distributions" and "priors". It would be helpful to define these terms to help readers who are not familiar with such terminology. Lines 145-147 could be clarified as follows: The input parameters in the simulation model described above require initial values to be assigned, along with their probability

distributions.  These prior distributions ('priors') are developed to estimate mole fractions, emissions, and banks for CFC-11, 12, 113, 114, and 115, HCFC-22, 141b, and 142b, and halon-1201, and 1311.

Thank you, this has been corrected.

2. Lines 187-190.  This sentence has a grammatical issue. "While there are published estimates of uncertainties in observed mole fractions, the uncertainties in modeled mole fractions do not, therefore, we estimate S separately for each chemical…"

Thank you, this has been corrected.

3. Table 3 conversions. I'm not sure how to interpret the units for GWP100 and OPD.  For the GWP100, is it Gg $CO_2$ equivalent per year?  For ODP, is it Gg CFC-11 equivalent per year?

Yes, GWP100 is weighting the amount of banked material by converting the banks into their warming potential relative of CO2 over a 100-year time horizon based on 391 ppm CO2 concentrations.  ODPs are with respect to CFC-11.  This has been further clarified in the table title.  ODPs from WMO 2018 also date back to WMO 2014 (see Velders and Daniel, 2014). They are semi-empirical ODPs computed from 1) fractional release values derived from observations of age-of-air and species concentrations, and 2) species lifetimes, most of which are from SPARC (2013) which were based on a combination of observations and models.

4. Disparities in CFC-115. Because the Bayesian model differs from the observed CFC-115 mole fractions, and the modeled emissions are very different from the observationally-derived emissions, how much confidence should we have regarding the magnitude of the bank estimates or emissions by source for this compound?

This is an excellent point.  Issues surrounding the modeling of CFC-115 are yet to be resolved.  Because CFC-115 comprises a relatively small portion of the bank, we have not explored this in greater detail in our analysis for the present manuscript.  However, we have added an additional point of discussion on CFC-115 non-dispersive usage in the discussion, Line 429 - 432:

"In addition, we do not account for non-dispersive production in our analysis, namely the production of chemicals as by-products.  It is possible, for example, that some of the discrepancies in CFC-115 emissions could be explained by non-dispersive emissions as identified by (Vollmer et al., 2018). "

5. Unexpected differences between Figures 3 and 4. For most compounds, I would expect the emission rate by source (Gg/yr, Fig 4) to be a fraction of the magnitude of the banks (Gg, Fig 3).  This is the case for CFC-11, CFC-12, HCFC-22, F141b, and F142b, all of which appear to have an emission rate of ~10-20% of the banks per year.   However, CFC-113 appears to have an emission rate that exceeds the entire bank size per year, CFC-114 appears to be 3 orders of magnitude larger, and F-115 appears to be 2 orders of magnitude larger.  Is that correct?  If so, does that mean that essentially all of those

compounds are dispersed immediately (i.e., that banks are inconsequential)?  Then why are CFC-115 emissions coming entirely from long-banks?  I seem to be missing something important here.

Thank you for bringing this to our attention.  I made an error plotting CFC-114 and 115 banks, this has been corrected.  CFC-113 however, was largely used as a solvent and thus not subject to significant banking.  In addition, 113 is used as a feedstock and emissions from feedstock use is possibly explaining this large source of emission.

6. Different time ranges on x-scale. This is a minor edit, but it would help the reader line up plots if the x-axes for the figures used the same time range.  Plots start in 1940, 1950, mid 1950s and 1960.

   We updated all figures to start in 1960 to 2020 for consistency.

7. In order to contextualize these results with prior studies, it might also be helpful to include the results from Lickley *et al*, 2020 and Lickley *et al*. 2021 on some of the plots.

   We added results for Lickley et al. 2020, which make the same atmospheric lifetime assumptions.

Overall, this is a useful extension of their prior studies, and the study will shine a light on the discrepancies between what is reported and what is happening in terms of these regulated halocarbons.  Given the importance, I think the above clarifications will help make the manuscript more accessible to a broader audience.